# Lung Dual-Energy CT Perfusion Blood Volume as a Marker of Severity in Chronic Thromboembolic Pulmonary Hypertension

**DOI:** 10.3390/diagnostics13040769

**Published:** 2023-02-17

**Authors:** Salim A. Si-Mohamed, Léa Zumbihl, Ségolène Turquier, Sara Boccalini, Jean-Francois Mornex, Philippe Douek, Vincent Cottin, Loic Boussel

**Affiliations:** 1Radiology Department, Louis Pradel Hospital, 59 Boulevard Pinel, 69500 Bron, France; 2INSA-Lyon, University of Lyon, Université Claude Bernard Lyon 1, UJM-Saint Etienne, CNRS, Inserm, CREATIS UMR 5220, U1206, 69621 Lyon, France; 3National Reference Center for Rare Pulmonary Diseases, Louis Pradel Hospital, Hospices Civils de Lyon, 69677 Lyon, France; 4UMR 754, INRAE, Claude Bernard University Lyon, 69007 Lyon, France; 5ERN-LUNG, 69500 Bron, France

**Keywords:** tomography, X-ray computed/methods, lung, perfusion, comparative study

## Abstract

In chronic thromboembolic pulmonary hypertension (CTEPH), assessment of severity requires right heart catheterization (RHC) through cardiac index (CI). Previous studies have shown that dual-energy CT allows a quantitative assessment of the lung perfusion blood volume (PBV). Therefore, the objective was to evaluate the quantitative PBV as a marker of severity in CTEPH. In the present study, thirty-three patients with CTEPH (22 women, 68.2 ± 14.8 years) were included from May 2017 to September 2021. Mean quantitative PBV was 7.6% ± 3.1 and correlated with CI (r = 0.519, *p* = 0.002). Mean qualitative PBV was 41.1 ± 13.4 and did not correlate with CI. Quantitative PBV AUC values were 0.795 (95% CI: 0.637–0.953, *p* = 0.013) for a CI ≥ 2 L/min/m^2^ and 0.752 (95% CI: 0.575–0.929, *p* = 0.020) for a CI ≥ 2.5 L/min/m^2^. In conclusion, quantitative lung PBV outperformed qualitative PBV for its correlation with the cardiac index and may be used as a non-invasive marker of severity in CTPEH patients.

## 1. Introduction

Chronic thromboembolic pulmonary hypertension (CTEPH) is a rare disease that may develop in the event of non-resolving pulmonary thromboemboli [1,2,3]. CTEPH is classified as a group 4 pulmonary hypertension (PH) in the current classification of the World Symposium on PH [4]. The precise pathogenesis is still unclear, but it is characterized by the presence of thromboembolic material in vessels and vascular remodeling [5,6,7,8]. Per current guidelines, non-invasive imaging modalities, V′/Q′ lung scintigraphy or single-photon emission CT (SPECT), are the recommended investigations in patients with PH to look for CTEPH [9,10]. However, assessment of severity still requires an invasive procedure such as right heart catheterization (RHC) through the assessment of the cardiac index in particular [5,9,11].

In the past two decades, dual-energy CT (DECT) systems have demonstrated new capabilities compared to conventional CT and allow the obtention of images specific to lung iodine content [12]. These images, which have been shown to be a surrogate marker of lung perfusion, enable the analysis of the lung perfusion blood volume (PBV) [13,14]. This PBV was shown to be correlated with qualitative and quantitative data provided by different lung perfusion nuclear methods using 99mTc macro aggregated albumin (MAA-Tc99m) such as multiplanar scintigraphy or cadmium-zinc-telluride camera SPECT-CT [15,16]. In CTEPH patients, the PBV was reported to be a marker of perfusion failure and correlated to hemodynamic parameters [11,17,18,19,20]. However, studies evaluating the PBV have been restricted so far, with flawed qualitative or semi-quantitative methods that are dependent on an observer’s experience, do not account for the absolute iodine concentration in the lungs, and do not allow it to be normalized to the concentration in the pulmonary trunk. Altogether, these limitations may limit the accuracy and impact of their analysis. Recently, the development of new software allows the quantification of the iodine concentration in pre-segmented structures such as lungs or pulmonary vessels, which may contribute to the improvement of PBV analysis relevance.

The objective of the present study was to evaluate the quantitative lung perfusion blood volume as a marker of severity in CTEPH.

## 2. Materials and Methods

### 2.1. Population

This single-center retrospective study in a reference center was approved by the local institutional review board (BLINDED, approval number 18-305). Written consent was waived due to the study’s retrospective character. Consecutive CTEPH patients who underwent DECT and RHC in both lungs between May 2017 and September 2021 were retrospectively analyzed. Patients for whom DECT and RHC examinations had been conducted more than 3 months apart were excluded. All included patients had a diagnosis of CTEPH confirmed by clinical history, examination, imaging (including CT, ventilation-perfusion scintigraphy or SPECT), RHC, and clinical follow-up. Patients who had non-exclusive postembolic hypertension were not included in the study.

### 2.2. Image Acquisition

DECT angiographies were performed on a dual-layer DECT system (iQon; Philips Healthcare^®^). The amount of iodinated contrast agent (iomeprol, iomeron 400 mg/mL; Bracco^®^) injected was calculated for each patient based on the time of acquisition for a flow rate of 3.5 mL/s using the following formula: volume of contrast agent = (estimated total time of CT acquisition + 6) * flow rate. The lung acquisitions were performed from cranial to caudal and CT scans were started using the bolus tracking technique with a threshold of 110 Hounsfield Units (HU) in the trunk of the main pulmonary artery. Conventional images and iodine maps were reconstructed for each patient using the Spectral Philips IntelliSpace Portal 12.0 (Philips HealthCare^®^) and stored for quantitative analysis. Time injection quality was evaluated by a ratio between the iodine concentration in the pulmonary trunk and in the left auricle, using an ROI sized at 150 mm^2^ on the iodine maps.

### 2.3. Quantitative Perfusion Blood Volume (PBV)

One radiologist (LZ, 3 years of experience) proceeded to a semi-automatic segmentation of both lungs using the IntelliSpace Portal software for conventional DECT images (COPD application; ISP 12, Philips Healthcare^®^). After segmentation of both lungs excluding the main pulmonary vessels, the mean iodine concentration for the two lungs was automatically measured from the iodine map for PBV calculation. As the time of injection and the iodine concentration in pulmonary vessels may impact the lung PBV [21], the quantitative PBV was normalized by quantifying the ratio between the mean iodine concentration of both lungs and the iodine concentration in a referent vessel. The referent vessel was defined by the trunk of the pulmonary artery (Figure 1). On average, this process takes 15 to 20 min.

### 2.4. Per-Segment Qualitative Analysis of the PBV

Two radiologists (LZ and SS-M, 3 and 7 years of experience) scored the extent of perfusion defects in each lung segment using the following scale [16]: 0: no defect; 1: defect in <25% of a segment; 2: defect in ≥25 and <50% of a segment; 3: defect in ≥50 and <75% of a segment; and 4: defect in ≥75% of a segment. The lung PBV score was the sum of the scores of the 18 segments analyzed, as previously published [20].

### 2.5. Assessment of Clinical Severity

Parameters measured by RHC examination included the systolic, diastolic, and mean pulmonary artery pressure (PAPs, PAPd, PAPm), right atrial pressure (RAP), cardiac output (CO), cardiac index (CI), and pulmonary vascular resistance (PVR). Brain natriuretic peptide (BNP), 6 min walk distance (6MWD), and the World Health Organization functional class (WHO fc) were recorded.

### 2.6. Statistical Analyses

Statistical analyses were performed with the IBM SPSS Statistics 22 (IBM, Armonk, NY, USA) and Prism software packages (version 8, GraphPad). Data are expressed as means ± mean standard errors and ranges (minimum–maximum), according to the normality tests (Shapiro–Wilk normality test). Pearson’s and Spearman’s correlation coefficients were calculated as a function of normality of variables’ distribution between normalized automated PBV, qualitative PBV, and the following parameters: mean iodine concentration in pulmonary artery, PAP, RAP, CO, CI, and PVR, WHO fc, 6MWD, BNP level. Correlations were described as <0.2 = very weak, 0.2 to 0.39 = weak, 0.40 to 0.59 = moderate, 0.60 to 0.79 = strong, and >0.8 = very strong. Data were categorized according to the CI thresholds (2 L/min/m^2^ and 2.5 L/min/m^2^), according to the PH prognosis determinants [9,22,23]. The areas under the curves (AUCs) for the receiver operating characteristic (ROC) analyses were calculated. The Youden’s method was used to determine the optimal threshold of PBV values and to calculate their sensitivity (Se) and specificity (Sp).

## 3. Results

### 3.1. Patients’ Characteristics

Thirty-three patients of median age 68 ± 15 (SD) years (range: 23–88) were included in the study (Table 1, Figure 2). The median delay between DECT and RHC was 5 days (IQR: 1–7), and it was ≤7 days for 26 patients (79%). All patients had anticoagulant treatment for more than 3 months at the time of the DECT and RHC. None of the patients underwent either surgical or endovascular intervention between DECT and RHC.

### 3.2. Correlation Analysis

Moderate to strong correlations were found between the quantitative PBV and the CI (r = 0.519; *p* = 0.002), CO (r = 0.672; *p* = 0.0001), and PVR (r = −0.466; *p* = 0.006), while no significant correlation was found between the qualitative PBV and the hemodynamics parameters (Table 2, Figure 3), despite a moderate correlation between quantitative and qualitative PBV (r = 0.41; *p* = 0.018).

### 3.3. Diagnostic Performances of Quantitative PBV for Cardiac Index Groups

An ROC analysis for CI ≥ 2 L/min/m^2^ produced an AUC value of 0.795 (95% CI: 0.637–0.953, *p* = 0.013) and for a CI ≥ 2.5 L/min/m^2^, an AUC value of 0.752 (95% CI: 0.575–0.929, *p* = 0.020). The PBV optimal cut-off values were 8% with a sensitivity of 52% and a specificity of 100% for a CI ≥ 2 L/min/m^2^, and 9.6% with a sensitivity of 54% and a specificity of 91% for a CI ≥ 2.5 L/min/m^2^ (Figure 4). Example cases are provided in Figure 5.

### 3.4. Radiation Dose Study

The mean volume CT dose index was 7.5 ± 3.4 (SD) mGy (range: 5.3–9.3), and the total dose length product was 283.6 ± 128.7 (SD) mGy.cm^−1^ (range: 187.5–343.5). As a result, the mean equivalent dose was calculated and found to be 3.9 ± 1.8 (SD) mSv (range: 2.6–4.8).

## 4. Discussion

In the present study, we demonstrated the additional value of having a full and quantitative assessment of the PBV by finding correlations between the quantitative PBV and the PVR, CI, and CO, in comparison to qualitative PBV. These findings are particularly interesting because of the prognostic value of these hemodynamic parameters commonly assessed during patient follow-up [9,22,23]. In addition, this allowed us to calculate optimal thresholds to differentiate the severity grades proposed by the international recommendations [9]. To our knowledge, the present study is the first to show such a correlation as well as diagnostic performances to assess the severity of CTEPH.

Numerous studies using DECT systems have highlighted the potential interest of PBV to indirectly evaluate RHC parameters in CTPEH patients, but with some limitations. A first study in 2010 by Hoey et al. demonstrated in a small cohort a strong correlation between qualitative PBV and mosaic attenuation pattern; however, no correlation with the vascular obstructive index, the mean pulmonary artery pressure, or the pulmonary vascular resistance was found [24]. A second study in 2016 by Takagi et al. found in a larger cohort study of 46 patients significant correlations between the PBV score, assessed in a semi-quantitative manner, and the PAP (mean, rho = 0.48; systolic, rho = 0.47; diastolic, rho = 0.39), PVR (rho = 0.47), and RVP (rho = 0.48) (all *p*-values < 0.01) [25]. Despite the fact that the population study did have comparable CI values to our population (i.e., 2.58 vs. 2.39 mL/min/m^2^), they did not find any correlation with CI. Nevertheless, the PAP values were much lower than in our study (24 mmHg vs. 41 mmHg in the present study), highly suggestive of differences between the study populations. In addition, the PBV calculation was limited to a semi-quantitative analysis using a score of perfusion defects per segment. A third study in 2013 by Meinel et al. reported a different process for calculating the PBV [26]. By quantifying the lung parenchyma attenuation, dividing it by the main pulmonary trunk enhancement, and using a calibration factor of 0.15, they calculated the PBV and reported significant correlations with the PAP, while no correlation was found between the PVR and the CI. In that study, despite the use of a DECT system, the iodine maps were not considered for PBV assessment, which may have limited its significance. A fourth study in 2020 by Tsutsumi et al. found a significant correlation between the PBV and the PAP, RAP, and PVR in 52 patients, while no correlation was found with the CI [27]. The authors extracted the PBV attenuation values but did not perform normalization with the pulmonary trunk enhancement. Furthermore, the injection protocol was different than the one used in our present study: A bolus tracking method was used with an ROI in the ascending aorta instead of an ROI located in the pulmonary trunk. This may have strongly impacted the PBV values because of a collateral circulation that may participate by 30% of the pulmonary inflow [28,29]. An injection at an earlier time would have been preferable to highlight a pulmonary vascular obstruction and limit the participation of systemic vascularization, such as suggested in other studies [11,30,31]. Lastly, a fifth study in 2022 by Kroeger et al. used an automated semi-quantitative volumetric process to estimate the PBV and found a correlation between the score of malperfused volume and the PVR but no correlation with the mPAP, results comparable to those of the present study [32]. Altogether, these studies support the existence of a relation between the degree of malperfusion in CTEPH and a hemodynamic impact, though with some discrepancies, and importantly without showing a correlation with the CI, despite the fact that it is known to play a key role in the severity assessment of CTEPH [9,11,22,23].

Our study has several limitations. Due to a limited number of patients and the absence of an external cohort, this study cannot be considered as a validation of PBV thresholds but only as a preliminary study. In addition, it is known that the streak artefacts due to the presence of iodine contrast agent in the superior vena cava and the subclavian vein could impact the quantitative PBV calculation. For example, Tsutsumi et al. used a lung PBV calculation that excluded the right upper zone to avoid artefacts [27]. To minimize this impact, the volume of contrast agent injected should be adjusted to the acquisition duration, such as performed in our clinical routine [16,31]. Finally, the investigation of a PBV value in control healthy subjects for a better understanding of its significance is missing. Other studies investigated the lung PBV in a population with and without vascular obstruction (especially in acute pulmonary embolism) [33], but no study has yet explored the normal value of lung PBV in a standard population in a quantitative manner, which leaves the question open to further investigations.

## 5. Conclusions

Quantitative lung perfusion blood volume outperformed the qualitative lung perfusion blood volume in terms of correlation with the hemodynamic parameters of CTPEH. The quantitative lung perfusion blood volume should be investigated as a non-invasive marker of severity to classify patients with a cardiac index ≥ 2 mL/min/m^2^ and ≥2.5 mL/min/m^2^.

## Figures and Tables

**Figure 1 diagnostics-13-00769-f001:**
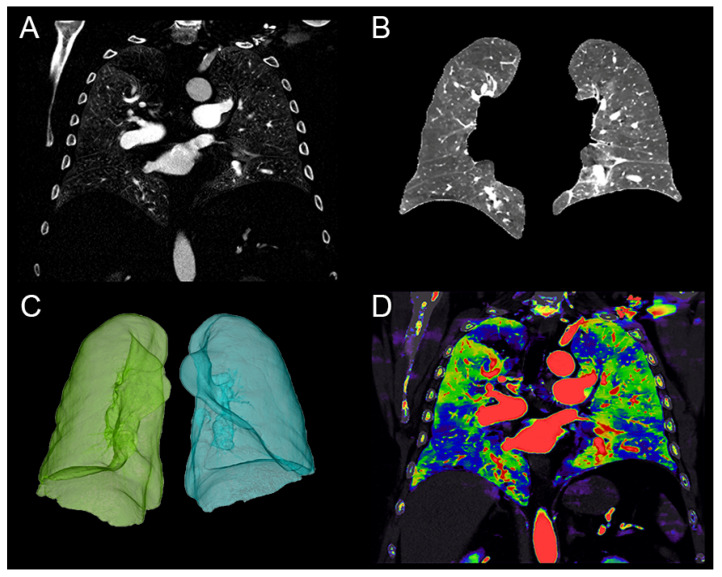
Quantification of the lung perfusion blood volume (PBV) in a 64-year-old male patient with chronic thromboembolic pulmonary hypertension (CTEPH). (**A**) Coronal image of the lung iodine density map. (**B**) Coronal image after lung segmentation with the semi-automatic software COPD (IntelliSpace Portal; Philips Healthcare^®^) allowing the exclusion of the main pulmonary vessels. (**C**) Volumetric segmentation of the lungs. (**D**) Overlay coronal image of the conventional and iodine density maps representing the defect perfusion of the CTEPH condition.

**Figure 2 diagnostics-13-00769-f002:**
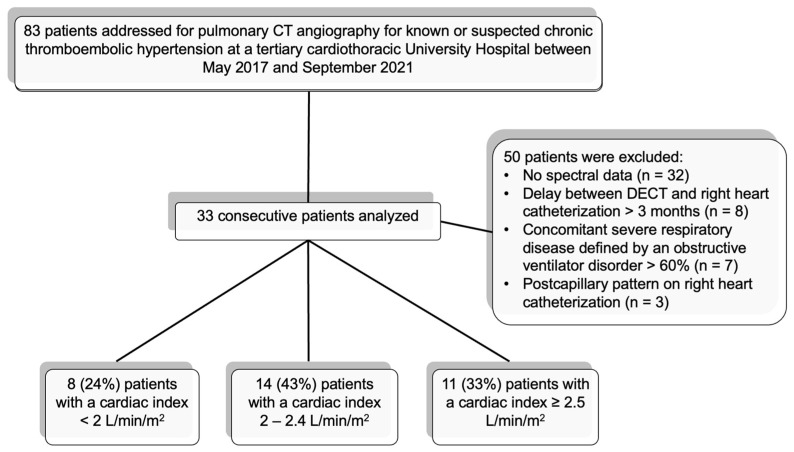
Flow chart of the study population.

**Figure 3 diagnostics-13-00769-f003:**
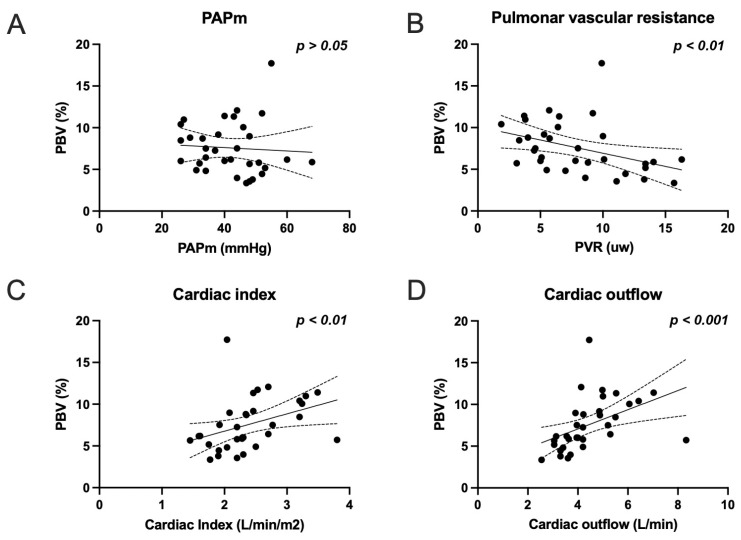
Correlation graphs between quantitative perfusion blood volume (PBV) and hemodynamics parameters; (**A**) PAPm (mean pulmonary arterial pressure); (**B**) PVR (pulmonary vascular resistance); (**C**) CI (cardiac index); and (**D**) CO (cardiac output).

**Figure 4 diagnostics-13-00769-f004:**
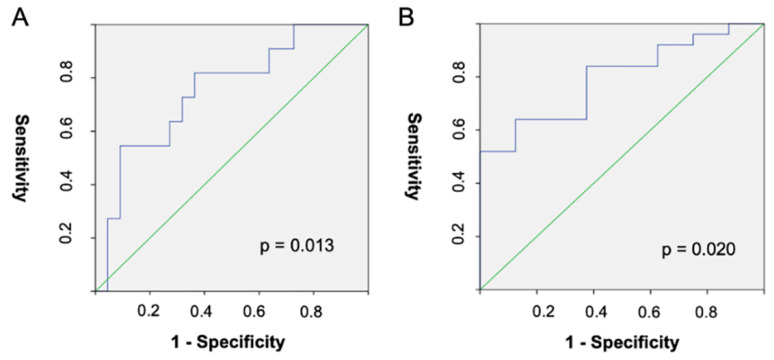
Receiver operating curves for quantitative PBV and cardiac index ≥ 2.5 L/min/m^2^ (**A**) or cardiac index ≥ 2 L/min/m^2^ (**B**).

**Figure 5 diagnostics-13-00769-f005:**
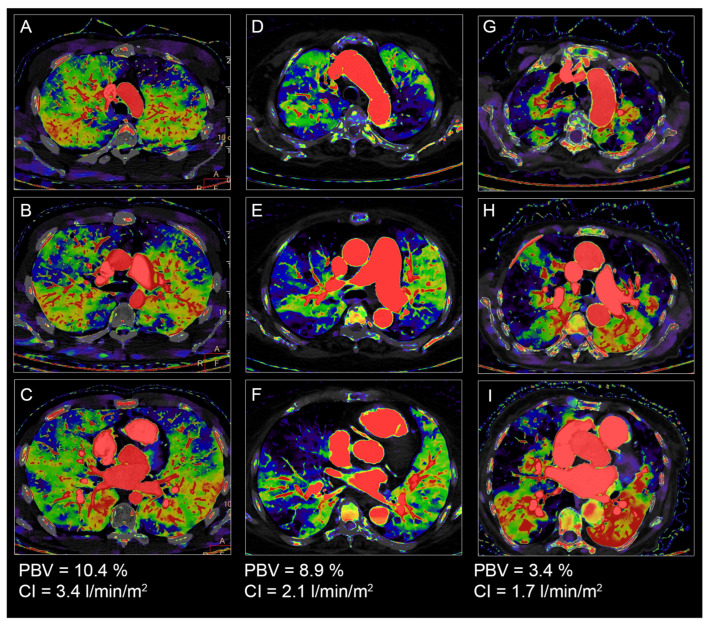
Cases of chronic thromboembolic pulmonary hypertension patients. (**A**–**C**). 23-year-old male patient with a quantitative lung perfusion blood volume (PBV) of 10.4% and a cardiac index of 3.4 L/min/m^2^. (**D**–**F**) 73-year-old female patient with a quantitative PBV of 8.9% and a cardiac index of 2.1 L/min/m^2^. (**G**–**I**) 86-year-old female patient with a quantitative PBV of 3.4% and a cardiac index of 1.7 L/min/m^2^.

**Table 1 diagnostics-13-00769-t001:** Patients’ characteristics.

Population (*n* = 33)	Value *	±SD	[Min–Max]
Age (years)	68	±15	[63–78]
Sex (male)	11 (33.3%)		
Height (cm)	165.2	±8.5	[150–183]
Weight (kg)	77.3	±16.7	[44–114]
Body mass index (kg/m^2^)	28.1	±5.1	[17–38]
Delay between pulmonary CT angiography and right heart catheterization	5		[1–7]
DECT parameters			
Ratio of the iodine concentrations in the pulmonary trunk and the left auricle	2.4		[1.7–3.2]
Quantitative perfused blood volume	6.5		[5.4–9.6]
Qualitative perfused blood volume	41.1	±13.4	[16–67]
Iodine concentration in the pulmonary trunk (mg/mL)	11.12		[9.16–13.72]
Right heart catheterization parameter			
Mean pulmonary artery pressure (mmHg)	41.8	±10.4	[26–68]
Systolic pulmonary artery pressure (mmHg)	70.9	±17.5	[39–106]
Diastolic pulmonary artery pressure (mmHg)	25.88	±8.2	[14–48]
Right atrial pressure (mmHg)	7.1	±3.4	[1–16]
Pulmonary capillary wedge pressure (mmHg)	9.7	±3.4	[5–18]
Pulmonary vascular resistance (WU)	7.7	±3.9	[1.9–16.3]
Cardiac output (L/min)	4.2		[3.6–5.1]
Cardiac index (L/min/m^2^)	2.4	±0.6	[1.4–3.8]
Other parameters			
6 min walk distance (min) (*n* = 28)	365	±164	[85–645]
Brain natriuretic peptide (ng/L)	171		[23.5–713]
WHO-fc (*n* = 32), *n* (%)			
Grade I	4 (12.5%)		
Grade II	14 (43.7%)		
Grade III	12 (37.5%)		
Grade IV	2 (6.3%)		

**Footnote.** SD: standard deviation, WHO fc: World Health Organization functional class. * Values are expressed as median [1st quartile–3rd quartile] or mean (±SD) [minimal–maximal], as appropriate.

**Table 2 diagnostics-13-00769-t002:** Correlation analysis between lung perfusion blood volume (PBV) and hemodynamics parameters. *Top row: Pearson’s coefficients, Bottom row: Spearman’s coefficients*.

		PAPs	PAPd	PAPm	RAP	PAWP	PVR	CO	CI	6MWD	BNP	WHOfc
Quantitative PBV	r	−0.302	−0.194	−0.181	0.103	0.118	−0.466 **	0.672 ***	0.519 **	0.071	−0.307	−0.189
	*p*	0.088	0.279	0.314	0.568	0.513	0.006	0.0001	0.002	0.696	0.083	0.291
Qualitative PBV	r	0.086	0.231	0.15	0.009	−0.081	0.234	−0.251	−0.095	0.089	0.339	0.188
	*p*	0.636	0.195	0.405	0.959	0.655	0.19	0.159	0.597	0.621	0.054	0.294

**Footnote.** PAPs: systolic pulmonary artery pressure, PAPd: diastolic pulmonary artery pressure, PAPm: mean pulmonary artery pressure, RAP: right atrial pressure, PAWP: pulmonary arterial wedge pressure, PVR: pulmonary vascular resistance, CO: cardiac output, CI: cardiac index, 6MWD: 6 min walk distance, BNP: brain natriuretic level, WHOfc: World Health Organization functional class, PBV: perfusion blood volume. ** *p*-value < 0.01, *** *p*-value < 0.001.

## Data Availability

Not applicable.

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
