# Peer review of "Lung Dual-Energy CT Perfusion Blood Volume as a Marker of Severity in Chronic Thromboembolic Pulmonary Hypertension"

_diagnostics, 2023, doi:10.3390/diagnostics13040769_

Round 1

Reviewer 1 Report

Introduction:

1. I found the goal confusing. It also does not represent what was done in the study. You did not compare quantitative vs. qualitative. Your focus was on quantitative but you also did on qualitative to prove your point. I would just say the purpose was to evaluate quantitative lung perfusion blood volumes a marker of severity in CTEPH and end there.

Methods

1. One radiologist (__, 3 years of experience) proceeded to a semi-automatic 82 segmentation of both lungs using the COPD for conventional DECT images (IntelliSpace 83 Portal; Philips Healthcare®). --> would recommend adding software in front of COPD, as it is confusing the way it is currently written.

Results/Discussion:

1. The main limitation of this manuscript is having a sample of 33 patients only and trying to establish a cutoff value for quantitative lung volume and drawing conclusions from it. You used two different CI points, which does not add value with a limited sample. I believe the best one could do with your dataset is to establish the most relevant CI based on literature, which correlates with being a good predictor of a hard outcome, such as 2.0 for example.  Provide evidence as why you chose this cutoff, otherwise, readers identify this was arbitrarily chosen to fit your data. Then I would come up with one QPBV cutoff point based on that but clearly acknowledge the limitation of what you are doing. ROC curves are not supposed to be drawn based on such small data. Your first paragraph of discussion and conclusion are not statistically appropriate due to the lack of a validation cohort. At least a second validation cohort to draw those conclusions, as the cutoff you found on your ROC curve has no meaningful value unless you are able to validate it on a separate cohort, ideally externally, but internally would be fine for a preliminary study.

2. Therefore, I suggest downgrading your conclusions as they are stronger than your results. The first sentence of conclusion is appropriate. The second sentence is too strong, you do not know yet if QPBV can be used to classify severity based on your study, as you have not validated this cutoffs, so please adjust accordingly.

Author Response

Reviewer 1:

Introduction:

  1. I found the goal confusing. It also does not represent what was done in the study. You did not compare quantitative vs. qualitative. Your focus was on quantitative but you also did on qualitative to prove your point. I would just say the purpose was to evaluate quantitative lung perfusion blood volumes a marker of severity in CTEPH and end there.

Answer: We thank te reviewer for this remark. We have corrected the objective accordingly.

Methods

  1. One radiologist (__, 3 years of experience) proceeded to a semi-automatic 82 segmentation of both lungs using the COPD for conventional DECT images (IntelliSpace 83 Portal; Philips Healthcare®). --> would recommend adding software in front of COPD, as it is confusing the way it is currently written.

Answer: Corrected.

Results/Discussion:

  1. The main limitation of this manuscript is having a sample of 33 patients only and trying to establish a cutoff value for quantitative lung volume and drawing conclusions from it. You used two different CI points, which does not add value with a limited sample. I believe the best one could do with your dataset is to establish the most relevant CI based on literature, which correlates with being a good predictor of a hard outcome, such as 2.0 for example. 

Answer: This can be found in the methods so it reads: "Data were categorized according to the CI thresholds (2 L/min/m2 and 2.5 L/min/m2), accordingly to the PH prognosis determinants [9, 22, 23]."

Provide evidence as why you chose this cutoff, otherwise, readers identify this was arbitrarily chosen to fit your data.

Answer: This can be found in the methods so it reads: "Data were categorized according to the CI thresholds (2 L/min/m2 and 2.5 L/min/m2), accordingly to the PH prognosis determinants [9, 22, 23]."

Then I would come up with one QPBV cutoff point based on that but clearly acknowledge the limitation of what you are doing. ROC curves are not supposed to be drawn based on such small data. Your first paragraph of discussion and conclusion are not statistically appropriate due to the lack of a validation cohort. At least a second validation cohort to draw those conclusions, as the cutoff you found on your ROC curve has no meaningful value unless you are able to validate it on a separate cohort, ideally externally, but internally would be fine for a preliminary study.

Answer: Corrected. We added in the limitation section the following limitation: "Due to a limited number of patients and the absence of an external cohort, this study can not be considered as a validation of PBV thresholds but only as a preliminary study."

  1. Therefore, I suggest downgrading your conclusions as they are stronger than your results. The first sentence of conclusion is appropriate. The second sentence is too strong, you do not know yet if QPBV can be used to classify severity based on your study, as you have not validated this cutoffs, so please adjust accordingly.

Answer: Corrected.

Reviewer 2 Report

The authors present the use of dual-energy CT imaging to assess pulmonary blood volume in the lungs of patients suffering from CTEPH both quantitatively and qualitatively. They also make use of correlations with well-known haemodynamic parameters associated with CTEPH in an attempt to benchmark which approach should be routinely used in this specific patient cohort.

Whilst the authors have clearly put much effort into the work and their results are indeed very interesting, there are certain aspects of the work, that in my opinion, require increased explanation. Please see below for more detailed comments.

Overall comments

·       The introduction would benefit from additional information on the limitations of semi-quantitative and qualitative measures of PBV

·       A reference to the amount of time required to generate qualitative vs quantitative PBV would be welcomed

·       The first two paragraphs of the discussion feel repetitive – I would suggest revising

Abstract

·       Data is reported in multiple ways – this makes it difficult to follow – I would suggest choosing one format and reporting all data in the same way

·       Units required for all metrics reported

·       Overall it is difficult to follow the abstract and I would suggest rewriting it

Figures / Tables

·       Figure 1 could be more detailed and more representative of a workflow rather than a collection of images showing each stage more clearly

·       Table 1 needs days adding to the delay (I realise it is in the paragraph above but just to be consistent as the rest of the table has units)

·       Table 1 should have a column for: meanSD and range rather than be grouped as it currently is

·       Tables 2 and 3 could be combined into a single table with r and p values in one row

·       Figure 3 – suggest adding correlation coefficient and p value to upper right corner of each

Specific comments

·       Page 1 line 39 – suggest adding ‘through the assessment of the cardiac index…’

·       Page 2 line 64 – any particular reason for 3 months?

·       Page 3 – line 113 – which normality tests were carried out?

Author Response

Reviewer 2:

The authors present the use of dual-energy CT imaging to assess pulmonary blood volume in the lungs of patients suffering from CTEPH both quantitatively and qualitatively. They also make use of correlations with well-known hemodynamic parameters associated with CTEPH in an attempt to benchmark which approach should be routinely used in this specific patient cohort.

Whilst the authors have clearly put much effort into the work and their results are indeed very interesting, there are certain aspects of the work, that in my opinion, require increased explanation. Please see below for more detailed comments.

Overall comments

  • The introduction would benefit from additional information on the limitations of semi-quantitative and qualitative measures of PBV

Answer: This has been corrected so it now reads: However, studies evaluating the PBV have yet been restricted with qualitative or semi-quantitative methods that are not flawless because they are dependent on an observer's experience, do not account for the absolute iodine concentration in the lungs, and do not allow to normalize it to the concentration in the pulmonary trunk. Altogether, these limitations may limit the accuracy and impact of their analysis.

  • A reference to the amount of time required to generate qualitative vs quantitative PBV would be welcomed.

Answer: This has been added in the methods so it now reads: On average, this process is taking 15 to 20 minutes.

  • The first two paragraphs of the discussion feel repetitive – I would suggest revising

Answer: We thank with the reviewer. We have deleted the first paragraph.

Abstract

  • Data is reported in multiple ways – this makes it difficult to follow – I would suggest choosing one format and reporting all data in the same way

Answer: we are sorry for this confusion. We have chosen to provide the mean ± SD for the values so it now reads: Mean quantitative PBV was of 7.6% ± 3.1 and correlated with CI (r=0.519, P=0.002). Mean qualitative PBV was of 41.1 ± 13.4 and did not correlate with CI. Quantitative PBV AUC were 0.79.

  • Units required for all metrics reported

Answer: Corrected. However qualitative PBV do not require units.

  • Overall, it is difficult to follow the abstract and I would suggest rewriting it

“ Answer: Corrected so it now reads: Abstract: In chronic thromboembolic pulmonary hypertension (CTEPH), assessment of severity requires right heart catheterization (RHC) through cardiac index (CI). Previous studies have shown that dual-energy CT allows a quantitative assessment of the lung perfusion blood volume (PBV). Therefore, the objective was to evaluate the quantitative PBV as a marker of severity in CTEPH. In the present study, thirty-three patients with CTEPH (22 women, 68.2 ± 14.8 years) were included from May 2017 to September 2021. Mean quantitative PBV was of 7.6% ± 3.1 and correlated with CI (r=0.519, P=0.002). Mean qualitative PBV was of 41.1 ± 13.4 and did not cor-relate with CI. Quantitative PBV AUC were 0.795 (95% CI: 0.637–0.953, P=0.013) for a CI ≥ 2 L/min/m2 and 0.752 (95% CI: 0.575–0.929, P=0.020) for a CI ≥ 2.5 L/min/m2. In conclusion, quantitative lung PBV outperformed qualitative PBV for its correlation with the cardiac index and may be used as a non-invasive marker of severity in CTPEH patients.”

Figures / Tables

  • Figure 1 could be more detailed and more representative of a workflow rather than a collection of images showing each stage more clearly.

Answer: We think that the current figure is fitting well with the analysis process and wish to keep it as it is.

  • Table 1 needs days adding to the delay (I realise it is in the paragraph above but just to be consistent as the rest of the table has units)

Answer: corrected.

  • Table 1 should have a column for: meanSD and range rather than be grouped as it currently is

Answer: corrected.

  • Tables 2 and 3 could be combined into a single table with r and p values in one row

Answer: corrected.

  • Figure 3 – suggest adding correlation coefficient and p value to upper right corner of each

Answer: corrected.

Specific comments

  • Page 1 line 39 – suggest adding ‘through the assessment of the cardiac index…’

Answer: Done.

  • Page 2 line 64 – any particular reason for 3 months?

Answer: It is an arbitrary choice based on previous literature.

  • Page 3 – line 113 – which normality tests were carried out?

Answer: A Shapiro-Wilk test was used. We have added it accordingly.

Reviewer 3 Report

This is a clinical study, which aimed to evaluate the quantitative lung perfusion blood volume as a marker of severity in CTEPH, compared with qualitative PBV. The authors indicated that quantitative lung perfusion blood volume outperformed the qualitative lung perfusion blood volume in terms of correlation with the hemodynamic parameters of CTPEH. The quantitative lung perfusion blood volume can be used as a non-invasive marker of severity to classify patients with a preserved cardiac index. This reviewer considers that the authors well performed the present study, and has only minor comments as described below. 

Major comment:

1.     Figures 1 should be bigger like Figure 5.

Minor comment:

2.     There were several typos. No name of radiologists in lines 82 and 99. “cardiax index” should be “cardiac index” in line 248. 

Author Response

Reviewer 3:

This is a clinical study, which aimed to evaluate the quantitative lung perfusion blood volume as a marker of severity in CTEPH, compared with qualitative PBV. The authors indicated that quantitative lung perfusion blood volume outperformed the qualitative lung perfusion blood volume in terms of correlation with the hemodynamic parameters of CTPEH. The quantitative lung perfusion blood volume can be used as a non-invasive marker of severity to classify patients with a preserved cardiac index. This reviewer considers that the authors well performed the present study, and has only minor comments as described below. 

Major comment:

  1. Figures 1 should be bigger like Figure 5.

Answer: corrected.

Minor comment:

  1. There were several typos. No name of radiologists in lines 82 and 99. “cardiax index” should be “cardiac index” in line 248. 

Answer: corrected.

Round 2

Reviewer 1 Report

Authors responded to my concerns appropriately. 

Reviewer 2 Report

Many thanks for addressing the points I raised previously. Whilst I still disagree on the format of figure 1 this is not a major sticking point and so I have no further comments.